# Compliance with the Updated BASHH Recurrent Vulvovaginal Candidiasis Guidelines Improves Patient Outcomes

**DOI:** 10.3390/jof8090924

**Published:** 2022-08-30

**Authors:** Lottie Brown, Mathilde Chamula, Sharon Weinberg, Frakinda Jbueen, Riina Rautemaa-Richardson

**Affiliations:** 1Division of Evolution, Infection and Genomics, Faculty of Biology, Medicine and Health, University of Manchester, Manchester M23 9LT, UK; 2Guys and St Thomas’ NHS Foundation Trust, London SE1 7EH, UK; 3Department of Infectious Diseases, Wythenshawe Hospital, Manchester University NHS Foundation Trust, Manchester M23 9LT, UK; 4Mycology Reference Centre Manchester, Wythenshawe Hospital, Manchester University NHS Foundation Trust, Manchester M23 9LT, UK; 5The Mid Yorkshire Hospitals NHS Trust, Wakefield WF1 4DG, UK

**Keywords:** recurrent vulvovaginal candidiasis, vulval skin care, vulval eczema, antifungal suppression, fluconazole, azole-resistance

## Abstract

Recurrent vulvovaginal candidiasis (RVVC) is a debilitating, chronic condition that affects over 138 million (6%) women of reproductive age annually. We performed a retrospective audit of RVVC referrals to our tertiary care *Candida* clinic to evaluate the impact of the significantly updated British Association of Sexual Health and HIV (BASHH) 2019 vulvovaginal candidiasis guidelines on patient outcomes, the principles of which were implemented at our centre at the onset of the guideline revision process in 2017. A total of 78 women referred with suspected RVVC in 2017–2020 were included. Their mean symptom duration prior to referral was 6.7 years. RVVC was the definitive diagnosis in 73% of cases. In the 27% of patients without RVVC, the most common diagnoses were acute VVC (29%), vulval eczema (14%), dry skin (14%) and vulvodynia (10%). Of those with RVVC, 60% were diagnosed with an additional diagnosis, most commonly vulval eczema or vulvodynia. Only 12% of women had been counselled on appropriate vulval skin care, the mainstay of RVVC management. Long-term antifungal suppression was initiated in 68% of women. Azole-resistant *Candida*, for which there is no licensed treatment available in the UK, was identified in 23% of women with RVVC. In the follow-up, 82% of patients reported good control of symptoms using antifungal suppression therapy and recommended skin care, 16% had partial symptom control with some “flare-ups” responding to treatment, none reported poor control and for 2% this information was not available. RVVC-related morbidity can be reduced by following the principles outlined in the BASHH guidelines.

## 1. Introduction

Vulvovaginal candidiasis (VVC) is an extremely common fungal infection of the lower female genital tract [1]. Between 5 and 10% of women of reproductive age will develop the recurrent disease, known as recurrent vulvovaginal candidiasis (RVVC) and defined as at least four episodes of VVC per year [1,2]. The source of VVC is the vaginal microbiome. Oestrogen is a driver of Candida growth, and a rise in this hormone at the onset of puberty induces Candida colonisation in the vagina. VVC is extremely rare in premenarchal girls and postmenopausal women off HRT, when oestrogen production diminishes. Symptomatic infection results from the host’s inflammatory response to exposure to *Candida* in tissue following superficial penetration of the mucosal lining via growth or mechanical rubbing [3]. For this reason, activities that cause friction on the vulval or vaginal skin, such as exercise or sexual intercourse, may contribute to the development of VVC and flare-up of RVCC. VVC or RVVC are not sexually transmitted despite this association. A number of other risk factors predispose women to RVVC, including poorly controlled diabetes mellitus, recent antibiotic use causing a disturbance in the vaginal flora, endogenous and exogenous oestrogen (including pregnancy, hormone replacement therapy (HRT) and the combined oral contraceptive pill), immunosuppression and iron, vitamin D or mannose binding lectin (MBL) deficiency [4,5,6,7]. MBL is a component of the innate immune defence against *Candida*, and deficiency occurs due to mutations in the MBL2 gene. A history of eczema, dry skin or behaviours, which may dry or irritate the delicate vulval and vaginal skin, including over-washing, use of feminine and other hygiene products, wearing sanitary pads or panty liners and douching, also predispose individuals to RVVC. In many cases, RVVC occurs in the absence of risk factors. A spectrum of *Candida* species can cause VVC, but the majority of cases are caused by *Candida albicans*, which are normally susceptible to all standard treatments. Repeated courses of antifungal treatment can lead to overgrowth of innately resistant species such as *C. glabrata* or resistant isolates of *C. albicans* [8,9].

The British Association for Sexual Health and HIV (BASHH) guidelines on the management of VVC for healthcare professionals working in departments offering specialist Genitourinary (GUM) services within the UK were updated in 2019 [2]. These guidelines are updated from those published in 2007 and offer recommendations on the diagnosis, treatment and prevention of both acute and recurrent VVC. Although the guidelines are aimed at GUM departments, the principles should be applied to other settings, such as family medicine (known as ‘primary care’ in the UK), using local care pathways. BASHH guidelines aimed specifically at family medicine clinicians were published in 2013 and offer largely similar recommendations for the diagnosis and management of RVVC [10].

VVC can be a diagnostic challenge. The symptoms are non-specific and many other infectious and non-infectious conditions present similarly. Differential diagnoses include vulval dermatitis or eczema, lichen sclerosus, chronic lichen simplex and vulvodynia, which is the most common chronic pain syndrome. Many women have dual pathology [11]. Bacterial vaginosis (BV) presents very differently from VVC and should not be considered a differential diagnosis. However, due to disturbance of the vaginal flora, some women may experience alternating episodes of VVC and BV. In cases of uncertainty, vaginal pH testing is a simple and inexpensive tool to differentiate episodes of VVC and BV [10]. However, it is important to keep in mind that in addition to VVC, the vaginal pH is also normal in patients with dermatological conditions such as vulval eczema. External genitalia should always be examined, in order to exclude alternative or co-existing vulval pathologies in RVVC [12]. To set the diagnosis of RVVC, at least two of the four or more episodes should be confirmed by microscopy or culture when the patient is symptomatic and at least one of these must be a culture [2]. If the response to treatment is partial or poor, full speciation and sensitivity testing is recommended on the fungal growth to detect intrinsically resistant species of *Candida* such as *C. krusei* and *C. glabrata* [13]. Furthermore, a significant proportion of patients are colonised with *Candida* species, which may not be contributing to their symptoms. Candida counts fluctuate across the menstrual cycle with changes in oestrogen levels, making interpretation of microbiology culture results challenging [14]. Therefore, the diagnosis of candidiasis has to build on the combination of symptoms and clinical findings, not culture results alone. Furthermore, if antifungal treatment results in the clearance of *Candida* (microbiological resolution) but the patient has persisting symptoms, it is unlikely that they are due to *Candida*, and other or additional causes must be considered.

Patient education on good vulval skin care is the mainstay of RVVC management. All patients should be encouraged to use an emollient (such as Doublebase^®^ or Dermol^®^) as a moisturiser, barrier cream and soap substitute daily [2]. Vulval dermatitis, primary or secondary, is commonly present in RVVC and predisposes women to further exacerbations [11,15]. The effect of emollient use is two-fold: Providing symptomatic relief and preventing further flare-ups. Patients should be advised to avoid anything that may dry and irritate the sensitive vulval skin, including using shower gels, wipes, soaps and feminine hygiene products, excessive cleaning of the vagina (including douching) and wearing daily panty liners and or tight, non-breathable fabrics [4,14,16,17,18]. For patients that report sexual activity triggering RVVC symptoms, a gentle water-based lubricant may be useful [2]. In addition to skincare education, antifungal therapy should be given as an induction regime and immediately followed by longer-term suppressive therapy, usually for at least 6 months, to achieve clinical remission [19,20]. After this time, if the patient is asymptomatic, therapy should be gradually withdrawn. Most patients will continue to have good control of symptoms but some may require additional periods of suppressive therapy for longer durations [2]. There is limited research on maintenance regimens, but one study found that by using an individualized reducing regimen of fluconazole, approximately 90% of women will remain disease-free at 6 months [21]. Antihistamines such as cetirizine may be offered in addition to suppression therapy, particularly if patients have a history of atopy [22]. Finally, nystatin and boric acid pessaries should be reserved for azole-resistant VVC and may be prescribed off-license at tertiary clinics in the UK [2]. There are currently no studies comparing the efficacy and tolerability of these drugs in RVVC. Key aspects of RVVC are summarized in Table 1.

An audit of RVVC referrals based on the BASHH guidelines has never been previously conducted. The aim of this audit was to evaluate the compliance of healthcare professionals referring to our Candida clinic with the current BASHH guideline. Additionally, we aimed to investigate the outcomes of patients attending the clinic, including definitive diagnosis, management and symptom control.

## 2. Materials and Methods

### 2.1. Data Collection

This retrospective audit was of patients referred to the tertiary *Candida* clinic at Wythenshawe Hospital, Manchester University NHS Trust. The *Candida* clinic is the only of its kind in the UK and has seen an increasing number of referrals for RVVC since its inception in 2012. Patients were referred from primary care (known in the UK as general practice) or secondary care (such as GUM, Gynaecology and Dermatology clinics). All patients referred to the clinic from 4 October 2017 to 1 July 2020 with suspected RVVC were included in the analyses. Electronic patient records were reviewed for patient demographics, referral origin, symptom duration, microbiological findings, known RVVC risk factors, final diagnosis, additional diagnoses, management plan and follow-up outcome. Individual case records were scrutinised and evaluated against auditable outcome measures (Table 2) outlined in the 2019 BASHH guidelines. The type and pattern of recorded symptoms were assessed against the typical clinical picture of RVVC, microbiology results, response to treatment and whether alternative differentials should have taken precedence. All patients were invited to complete the General Medicine Council (GMC) patient questionnaire 12 months after completion of the audit cycle. The questionnaire predominantly focuses on patients’ experience of our service and is separated into domains including the doctor–patient relationship, the explanation of condition and treatment, patient involvement in decision-making and treatment provided.

### 2.2. Statistical Analysis

Statistical analysis was performed using Microsoft Excel^®^ and GraphPad Prism^®^ version 9.0 (GraphPad Software Inc., San Diego, CA, USA). Continuous and categorical data on participant characteristics were presented as the median (range) and n (%), respectively. The Mann-Whitney U test and Fisher’s Exact Test were used to compare continuous and categorical data, respectively. A *p*-value below 0.05 was considered statistically significant.

### 2.3. Ethics

The NHS Health Research Authority (HRA) decision tools [23] were used to determine that our study was defined by the UK Policy Framework for Health and Social Care Research as an audit, not research, and that it did not require ethical committee approval. The audit was registered with the Manchester University NHS Foundation Trust Audit Department. Confidentiality of the patients was ensured by strictly excluding the collection of identifiable data.

## 3. Results

A total of 83 women were referred to the *Candida* clinic with suspected RVVC during the study period. Of these, 5 were excluded due to missing data and 78 were included for analysis. Patient demographics, diagnoses and outcomes are summarised in Table 3. Age at the time of referral ranged from 17–76 years with a mean age of 38 years. Symptom duration prior to referral ranged from 18 months to 21 years, with a mean duration of 6.7 years. RVVC was the definitive diagnosis in 57 patients (73%), with 34 patients (60%) also having a secondary diagnosis of bacterial vaginosis (BV) in 18% of cases (n = 10), vulvodynia in 18% of cases (n = 10) and vulval eczema in 16% of cases (n = 9). In 21 patients where RVVC was not the definitive diagnosis, primary diagnoses included acute VVC (29%), vulval eczema (14%) and dry skin (14%).

Patients referred from primary or secondary care were appropriately investigated prior to referral with a high vaginal swab (HVS) microscopy and/or culture in 87% and 88% of cases, respectively. Appropriate antifungal regimens, including long-term suppression, were trialled prior to referral in 69% of cases referred by primary-care doctors and in 66% of cases referred by secondary-care doctors. Only 16% of patients referred from secondary care and 7% of patients referred from primary care were educated on appropriate vulval skin care before attending clinic. Excessive or otherwise inappropriate personal hygiene likely to irritate vulval skin, including the use of soap and shower gel, was reported in 20% of cases.

Hormonal changes (in menarche, menopause and pregnancy) were reported as a catalyst for the development of symptoms in 26% of patients with RVVC and 19% with an alternative diagnosis (non-significant difference). Antibiotics were implicated as a catalyst in 23% of patients with RVVC and 19% with alternative diagnoses (non-significant difference). The use of hormonal agents, including HRT, the oral contraceptive pill, topical oestrogen, hormonal IUD and an implant were associated with symptoms in 30% of women with RVVC and 29% of women with another diagnosis. Symptoms of discharge and an itch were the most common complaint among women referred to the *Candida* clinic. Of patients with RVVC, 79% reported symptoms of discharge and 75% with an itch. Soreness (50%), burning (25%), pain during intercourse (21%) and dryness (14%) were also common symptoms experienced. There was no significant difference in the rates of reported symptoms in patients diagnosed with RVVC and those with alternative diagnoses.

Investigations at the *Candida* clinic identified *C. albicans* in 46% of women by the culture of HVS samples. The most common non-*albicans Candida* species was *C. glabrata*, which was reported in 12% of RVVC cases. Cases of RVVC due to *C. glabrata* were more common among older women (median age 44 years, compared to 39 years for other species) and associated with recent antibiotic use (33% vs. 20%). Close to 90% of *C. glabrata* cases were fluconazole-resistant. Other causative species included *C. parapsilosis*, *C. guielliermondii*, *C. krusei*, *C. lusitaniae*, *C dubliniensis*, *C. tropicalis* and *Pichia manshurica.* Mixed growth (2 or more yeast species) was identified in 11% of women. Fully susceptible species of *Candida* were isolated in 54% of women with RVVC and azole-resistant species in 24%. There was no growth in 19% of cases despite clear clinical indications of RVVC. MBL deficiency was significantly more common in patients with RVVC (42%) compared to those with alternative diagnoses (10%) (*p* = 0.0069).

Vulval skincare advice was given to all 78 women referred to the *Candida* clinic with suspected RVVC. The use of vaginal oestrogen cream was recommended to 12% of patients, all menopausal. Regular antihistamines were advised in 13% of cases, in patients with a history of atopy. All 57 patients diagnosed with RVVC were initiated on antifungal suppression. The choice of antifungal suppression therapy was based on implicated *Candida* species, a sensitivity profile and prior treatments trialled. Fluconazole suppression was used most commonly (in 58% of patients), followed by nystatin in 47%, boric acid in 23%, clotrimazole in 14% and itraconazole in 7%. More than one type of antifungal was trialled in 37% of RVVC patients. Long-term antifungal combination therapy (≥6 months) was initiated in 7% of cases. Off-license antifungals, including nystatin and boric acid pessaries, were required in 56% of women. The patient-reported response to treatment was recorded at their most recent follow-up appointment. Of the patients diagnosed with RVVC, 82% of follow-up clinic letters indicated good control of symptoms using the prescribed therapy and personal hygiene regime, 16% had partial symptom control and indicated ongoing “flare-ups”, no patients reported poor control and, for 2%, this information was not available. In patients with an alternative diagnosis, 71% indicated “good” control, 24% indicated partial control and 10% had poor control.

Of the 78 patients treated for suspected RVVC at the clinic, 38 completed the GMC patient questionnaire, yielding a response rate of 49%. All patients rated their clinic visit as ‘very important’ or ‘important’ to their health and wellbeing. A response of ‘very good or ‘good’ was recorded in 97% of respondents in the domains of doctor–patient relationship (politeness, ease during consultation, listening skills, maintenance of privacy and trust). There was a similarly positive response of 97% to the assessment of the medical condition, explanation of condition and treatment, involvement of the patient in decision-making and treatment provision.

## 4. Discussion

Recurrent vulvovaginal candidiasis is more of a diagnostic challenge than is often anticipated. In the present study, in nearly one-third of the cases, RVVC was ruled out as the cause of the patient’s symptoms explaining their poor clinical response to antifungal therapy. Furthermore, RVVC is often complicated by co-existing vulval pathologies [2,24]. Over half of the women diagnosed with RVVC in our study also had a secondary diagnosis. Importantly, the results of our study show that RVVC-related morbidity can be reduced by following the principles outlined in the BASHH guidelines and by improving the recognition and understanding of the condition in primary and secondary care. Key learning points from our study are summarised in Table 4.

Most women treated at our centre had been investigated by HVS microscopy and/or culture and trialled on appropriate azole antifungal regimens prior to referral, in accordance with the guidelines. Proven or possible azole-resistant infection was a common trigger for referral to our centre. In the UK, there are no licensed drugs for the treatment of azole-resistant VVC, and off-license nystatin and boric acid pessaries are only available in specialized centres. Over half of our patients were treated with these off-license therapies. With an estimated 100,000 women in the UK living with RVVC and even more affected by acute VVC, the demand for specialist services that can care for women with azole-resistant VVC is far too high for the few specialized services in the country [1]. There is a clear need for better access to effective treatment for microbiologically proven azole-resistant VVC.

Education of good vulval skin care is the mainstay of RVVC management and prevention but primary- and secondary-care clinicians are not trained in providing this. Concerningly, hygiene practices likely to irritate vulval skin were mentioned in 20% of cases, and only 12% of women had been previously counselled on appropriate personal hygiene. Most patients had had multiple consultations with healthcare professionals regarding their vulval symptoms prior to referral, and yet, a large majority were unaware of the importance of skincare in the prevention and relief of RVVC symptoms. The previous 2007 BASHH guidelines focused on antifungal management and vulval skincare were discussed only briefly. Poor awareness of the importance of vulval skincare may reflect the lack of emphasis placed on this aspect of RVVC care in previous guidelines.

Traditional, culture-based prevalence studies demonstrated *Candida* colonisation in 20% of women of reproductive age [25,26]. A more recent study using next-generation sequencing to characterise the vaginal mycobiome suggests a higher proportion of women are colonised. Among vaginal samples from 251 healthy, pre-menopausal women, *Candida* was identified in 69.9% [27]. Therefore, a vaginal swab positive for *Candida* is not diagnostic of VVC on its own. A diagnostic test that could distinguish between colonisation and infection would be an invaluable tool in both clinical practice and medical research. A recent phase III trial of a novel, non-azole antifungal (Ibrexafungerp) achieved clinical cure in only 50.5% of patients (compared to 28.6% with placebo) [28]. It is possible that the rate of clinical cure was reduced by the inclusion of patients with *Candida* colonisation and symptomatic due to other vulval pathologies. If the patient does not respond to microbiologically effective therapy, clinicians must consider alternative and additional diagnoses. In cases of dual pathology, treating VVC with antifungals without treating the inflammation of the vulval skin provides only partial relief to their symptoms and leaves them susceptible to reinfection and recurrence of symptoms. In our study, the most common secondary diagnoses were vulval eczema and vulvodynia. Vulval eczema is also primarily treated with emollients, which reiterates the importance of vulval skincare education in patients with this clinical picture [29]. In fact, the 2016 European guidelines for the management of vulval conditions recommend the use of emollients in all diseases affecting the area and this must be communicated to patients [30].

Bacterial vaginosis (BV) was another common secondary diagnosis. Younger patients may cycle between VVC and BV, but these two conditions clinically present very differently. Therefore, patient education is key so that they can tell the two apart and seek effective treatment. Bacterial vaginitis, on the other hand, is more common in post-menopausal, post-oestrogenic women off HRT following the increase in vaginal pH to neutral impacting the vulvovaginal flora, whilst the risk for VVC decreases after menopause as oestrogen is a driver of RVVC symptoms [14]. However, post-menopausal vulvovaginal atrophy is a known risk factor for RVVC. Our study found that elevated oestrogen levels, both exogenous (including HRT) and endogenous, were a catalyst for RVVC symptoms in nearly one-third of patients. Despite this, the use of vaginal oestrogen cream was recommended to post-menopausal patients with vulvovaginal atrophy, and all of these patients reported improved control of their symptoms.

In our patient group, the rate of MBL deficiency was over 3 times higher in those diagnosed with RVVC compared to those with an alternative primary diagnosis (55% compared to 15%). MBL deficiency is a congenital deficiency of the innate immune system for which there is no licensed treatment [31]. The main rationale for looking for MBL deficiency in these patients is that it can, in part, explain their RVVC and help to dispel harmful misconceptions around the cause of their condition, such as being “dirty”. It also helps to reassure patients that various *Candida* diets and nutritional supplements will not treat their RVVC.

There are some limitations to our study. Firstly, the sample size is limited by the single-centre design. However, this is the first comprehensive analysis of the impact of the updated BASHH guideline principles on patient outcomes. Secondly, as a retrospective study, the data collection depended on documentation in patient records. Medical notes for five patients were unavailable and they were excluded. Patients who were unresponsive to azole therapy (usually due to infection with intrinsically resistant *Candida*) were almost certainly over-represented in our study as they cannot be managed in primary care in the UK. Examination of the external genitalia in all women with suspected RVVC is an auditable outcome in BASHH guidelines. As there were no facilities to perform gynaecological examinations at the *Candida* clinic, a recent examination with recorded findings was a prerequisite for referral acceptance. Clinicians were potentially prompted to perform an examination as part of the referral process.

## 5. Conclusions

Our audit confirmed that the diagnosis of VVC and RVVC is more challenging than often thought. It is not rare that RVVC is complicated by coexisting vulval pathologies and symptoms labelled as RVVC are due to alternative diagnoses. Patient education on vulval skincare is the mainstay of RVVC management and prevention, but healthcare professionals are not trained to do this. After often years of suffering, all patients reported good or partial responses to the treatment at the follow-up. RVVC-related morbidity can be reduced by following the principles outlined in the BASHH guidelines.

## Figures and Tables

**Table 1 jof-08-00924-t001:** What is known about recurrent vulvovaginal candidiasis (RVVC).

RVVC is an extremely common condition affecting between 5–10% of women of reproductive age Diagnosing RVVC is challenging as a high proportion of women are colonised with Candida, the symptoms are not pathognomonic and there is significant overlap with other vulval pathologies such as vulval dermatitis and vulvodynia Practices which dry or irritate the delicate vulval skin, including over-washing, use of feminine and other hygiene products, wearing sanitary pads or panty liners, predispose women to RVVC. Instead, women should be advised to use an emollient as a moisturiser, barrier cream and soap substitute daily

**Table 2 jof-08-00924-t002:** Recommended auditable outcomes for management of RVVC [2]. Adapted with permission from the lead author Dr Cara Saxon on the behalf of the guideline writing group.

Auditable Outcomes	Performance Standard
All women with RVVC to be offered a genital examination performed by an appropriately trained clinician	90%
Microscopy and/or culture with speciation and sensitivity testing to be performed in all cases of suspected RVVC	90%
Discussion around offer or suppressive or alternative long term therapy for all women with proven RVVC to be documented	90%
Discussion around what constitutes good vulval skincare for all women with RVVC	90%

**Table 3 jof-08-00924-t003:** Patient characteristics, diagnoses and outcomes.

	RVVC	Non-RVVC	Total	*p* Value
Primary diagnosis, n (%)	57 (73)	21 (28)	78 (100)	-
Median age (range)	39 (20–73)	29 (17–76)	37 (17–76)	0.0754
Catalyst, n (%)				
Antibiotic	13 (11)	4 (20)	17 (23)	>0.9999
Menarche	4 (7)	2 (10)	6 (8)	0.6577
Menopause	5 (9)	0 (0)	5 (6)	0.3160
Pregnancy	6 (22)	2 (10)	8 (10)	>0.9999
Use of hormonal agents, n (%)	17 (30)	6 (29)	23 (29)	>0.9999
Symptoms, n (%)				
Discharge	44 (79)	13 (62)	57 (73)	0.2493
Itch	42 (75)	13 (62)	55 (71)	0.4022
Soreness	28 (50)	7 (33)	35 (45)	0.3052
Burning	14 (25)	3 (14)	17 (22)	0.5369
Dyspareunia	12 (21)	5 (24)	17 (22)	0.7664
Dryness	8 (14)	3 (14)	9 (12)	>0.9999
Secondary diagnoses, n (%)				
Bacterial vaginosis	10 (18)	4 (19)	14 (18)	-
Vulvodynia	10 (18)	6 (29)	16 (21)	-
Vulval eczema/dermatitis	9 (16)	7 (33)	16 (21)	-
Other	5 (9)	3 (14)	8 (10)	-
MBL deficiency, n (%) *	24 (55)	2 (15)	26 (33)	0.0069
HVS culture, n (%)				
Fully susceptible	31 (54)	2 (10)	33 (42)	-
Azole-resistant	13 (23)	2 (10)	15 (19)	-
No growth	12 (21)	17 (80)	29 (37)	-
Result not available	2 (1)	0 (0)	2 (3)	-
Management in clinic, n (%)				
Skin care advice	57 (100)	21 (100)	78 (100)	-
Topical oestrogen	6 (11)	1 (5)	7 (9)	-
Topical steroids	5 (9)	3 (14)	8 (10)	-
Antihistamine	5 (9)	5 (24)	10 (13)	-
Antifungal therapy prescribed at clinic, n (%)				
Acute	0 (0)	9 (43)	9 (12)	-
Single-agent suppression	53 (93)	2 (10)	55 (71)	-
Combination suppression	4 (7)	0 (0)	4 (5)	-
Off-license	32 (56)	10 (47)	42 (54)	-
Clinical response, n (%)				
Good	47 (82)	12 (57)	62 (79)	-
Partial	9 (16)	5 (24)	14 (18)	-
Poor	0 (0)	2 (10)	2 (3)	-
Not available	1 (2)	2 (10)	3 (4)	-

* Test performed by ELISA at the Immunology Department of the Great Ormond Street Hospital in London, UK.

**Table 4 jof-08-00924-t004:** Learning points from our study.

Co-existing vulval pathologies including vulval eczema and vulvodynia were common among our RVVC patients Diagnosis of RVVC should be based on a combination of clinical and microbiological examination, including careful history taking around triggers and response to antifungal therapy Vulval skin care is the mainstay of RVVC prevention and management but there is poor awareness of its importance among healthcare professionals in primary and secondary care and a high proportion of women reported harmful hygiene practices prior to referral. We recommend a personalized approach to patient education and treatment, based on individual triggers and risk factors. Over half of patients in our study were treated with off-license therapies which they would not otherwise had access to in primary care The study provides evidence that by following BASHH guideline on appropriate skin care and antifungal suppression, women can achieve good control of their symptoms

## Data Availability

All data relevant to the study are included in the article.

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
