# Peer review of "Compliance with the Updated BASHH Recurrent Vulvovaginal Candidiasis Guidelines Improves Patient Outcomes"

_jof, 2022, doi:10.3390/jof8090924_

Round 1
Reviewer 1 Report
General comment:
The results indicate the effect of age and hormonal status (menopause) on the frequency of RVVC vs. non-RVVC. Both factors (together with DM) also play a role in the occurrence of C. glabrata and could be more comprehensively commented (maybe as a specific subgroup of RVVC) in the discussion (what was the average age for patients with C. glabrata, what was its sensitivity to antifungals, any other predispositions - diabetes mellitus?) (Holland J. et al. 2003, Kennedy MA and Sobel JD 2010).
Specific comments:
Line 53: comma after immunosuppression
Line 87: There is no mention of vaginal pH (this also applies to the discussion – except of line 272). It is an important marker (easily and inexpensively implemented) for differential diagnosis of vaginal discharge. Was vaginal pH measured? The Methods mentions only the microbiological results/ findings in general.
Line 196: A closer specification of MBL (MBL2?) and methodology (in the Methods) for its determination are missing.
Line 252: Somewhat outdated citations 25 and 26 (follow from conventional culture-dependent techniques). More recent papers suggest a higher proportion of colonization of the vagina by yeasts. In this context, a brief note on the study of the vaginal mycobiome were appropriate (e.g., Drell T et al. 2013).
Line 273: This is true in general, but in reality, today many women are on hormone replacement therapy, which is probably behind the relatively higher yeast colonization and/ or incidence of R/VVC in older women (e.g., Dennerstein GJ and Ellis DH 2001, Devillar E. et al. 2004, Fisher G. and Bradford J. 2011).
Line 283: This may not be true; there are experimental studies of MBL administration to MBL-deficient patients although not to those with RVVC (Keizer MP at al. 2014).
Line 288:(Table 2): I would also emphasize a personalized approach, especially for treatment, based on a comprehensive diagnosis consists of clinical and microbiological examination (microscopy, vaginal pH, Whiff test), including a careful anamnesis.
Author Response
RE: Compliance with the updated BASHH recurrent vulvovaginal candidiasis guidelines improves patient outcomes (JoF 1837914)
Reviewer 1:
The results indicate the effect of age and hormonal status (menopause) on the frequency of RVVC vs. non-RVVC. Both factors (together with DM) also play a role in the occurrence of C. glabrata and could be more comprehensively commented (maybe as a specific subgroup of RVVC) in the discussion (what was the average age for patients with C. glabrata, what was its sensitivity to antifungals, any other predispositions - diabetes mellitus?) (Holland J. et al. 2003, Kennedy MA and Sobel JD 2010).
- Added subgroup analysis for glabrata in results: Cases of RVVC due to C. glabrata were more common among older women (median age 44 years, compared to 39 years for other species) and associated with recent antibiotic use (33% vs 20%). Close to 90% of C. glabrata cases were fluconazole resistant.
Line 53: comma after immunosuppression
- Noted – many thanks.
Line 87: There is no mention of vaginal pH (this also applies to the discussion – except of line 272). It is an important marker (easily and inexpensively implemented) for differential diagnosis of vaginal discharge. Was vaginal pH measured? The Methods mentions only the microbiological results/ findings in general.
Measuring vaginal pH is not recommended in the BASHH guideline. This is mainly because it is best at differentiating VVC from BV (normal vs high pH) but BV presents clinically so differently (vaginitis vs vaginosis, level of skin involvement) that it is not even listed as one of the main differentials in the guideline. Also, women with Candida vaginitis typically have a normal vaginal pH as so do those with vulval eczema and lichen sclerosus. Therefore, the normal pH result can be misleading and lead to overuse of antifungals. That said, vaginal pH testing can be helpful in patients who flip between VVC and BV. We have modified the text as below:
- Introduction: Bacterial vaginosis (BV) presents very differently to VVC and should not be considered a differential diagnosis. However, due to disturbance of the vaginal flora, some women may experience alternating episodes of VVC and BV. In cases of uncertainty, vaginal pH testing is a simple and inexpensive tool to differentiate episodes of VVC and BV [10]. However, it is important to keep in mind that in addition to VVC, the vaginal pH is normal also in patients with dermatological conditions such as vulval eczema.
Line 196: A closer specification of MBL (MBL2?) and methodology (in the Methods) for its determination are missing.
- Added to background: MBL is a component of the innate immune defence against Candida and deficiency occurs due to mutations in the MBL2 gene.
- This is a ‘send away’ test and not performed locally. Added as a footnote in Table 2: Test performed by ELISA at the Immunology Department of the Great Ormond Street Hospital in London, UK.
- http://www.labs.gosh.nhs.uk/laboratory-services/immunology/tests/mannose-binding-lectin
Line 252: Somewhat outdated citations 25 and 26 (follow from conventional culture-dependent techniques). More recent papers suggest a higher proportion of colonization of the vagina by yeasts. In this context, a brief note on the study of the vaginal mycobiome were appropriate (e.g., Drell T et al. 2013).
- Added: Traditional, culture-based prevalence studies demonstrated Candida colonization in 20% of women of reproductive age [25,26]. A more recent study using next-generation sequencing to characterise the vaginal mycobiome suggests a higher proportion of women are colonized. Among vaginal samples from 251 healthy, pre-menopausal women, Candida was identified in 69.9% [27].
Line 273: This is true in general, but in reality, today many women are on hormone replacement therapy, which is probably behind the relatively higher yeast colonization and/ or incidence of R/VVC in older women (e.g., Dennerstein GJ and Ellis DH 2001, Devillar E. et al. 2004, Fisher G. and Bradford J. 2011).
- We fully agree and the following paragraph has been modified to emphasise this better: Bacterial vaginitis, on the other hand, is more common in post-menopausal, post-oestrogenic women off HRT following the increase of vaginal pH to neutral impacting the vulvovaginal flora, whilst the risk for VVC decreases after the menopause as oestrogen is a driver of RVVC symptoms [14]. However, post-menopausal vulvovaginal atrophy is a known risk factor for RVVC. Our study found that elevated oestrogen levels, both exogenous (including HRT) and endogenous, were a catalyst for RVVC symptoms in nearly a third of patients. Despite this, the use of vaginal oestrogen cream was recommended to post-menopausal patients with vulvovaginal atrophy, and all of these reported improved control of their symptoms.
Line 283: This may not be true; there are experimental studies of MBL administration to MBL-deficient patients although not to those with RVVC (Keizer MP at al. 2014).
- Added: MBL deficiency is a congenital deficiency of the innate immune system for which there is no licensed
Line 288:(Table 2): I would also emphasize a personalized approach, especially for treatment, based on a comprehensive diagnosis consists of clinical and microbiological examination (microscopy, vaginal pH, Whiff test), including a careful anamnesis.
- Added to learning points: Diagnosis of RVVC should be based on a combination of clinical and microbiological examination, including careful history taking around triggers and response to antifungal therapy. We recommend a personalized approach to patient education and treatment, based on individual triggers and risk factors.
Reviewer 2 Report
This review addresses adherence to new guidelines to adderss RVCC. It nicely demonstrates that the addition of recommended skin care is highly helpful with treating RVCC. Even with the small numbers, this finding is of interest. . Thus hopefully this study will encourage all health care facilities to educate patients in vulval skin care. The study also shows that it is important to deterimne if the patient is actually suffering from a Candida infetion opposed to other syndromes in which symptoms are shared.
The only point that isn't quire clear is the treatments of the study participants. In Table 2, assumedly antifungal therapy refers to the type of antifungal therapy the patients got prior to the study? The numbers in line 172 are different. Can this be clarified?
The follow up treatment is also not clear. If all the participants were seen in the same facility for this study, why were they prescribed different antifungal regimens? Could this be clarified?
Overall the manuscript summarizes what a specialized facility has accomplished by following new RVVC guidelines. It seems that these guidelines could be followed in primary care and other healthcare facilities.
Author Response
RE: Compliance with the updated BASHH recurrent vulvovaginal candidiasis guidelines improves patient outcomes (JoF 1837914)
Reviewer 2:
This review addresses adherence to new guidelines to address RVCC. It nicely demonstrates that the addition of recommended skin care is highly helpful with treating RVCC. Even with the small numbers, this finding is of interest. Thus hopefully this study will encourage all health care facilities to educate patients in vulval skin care. The study also shows that it is important to determine if the patient is actually suffering from a Candida infection opposed to other syndromes in which symptoms are shared.
The only point that isn't quite clear is the treatments of the study participants. In Table 2, assumedly antifungal therapy refers to the type of antifungal therapy the patients got prior to the study? The numbers in line 172 are different. Can this be clarified?
- Table 2 refers to antifungal therapy prescribed at clinic. Line 172 refers to antifungal therapy prior to referral.
- Table 2 amended to make this clearer.
The follow up treatment is also not clear. If all the participants were seen in the same facility for this study, why were they prescribed different antifungal regimens? Could this be clarified?
- Please see results: All 57 patients diagnosed with RVVC were initiated on antifungal suppression. Choice of antifungal suppression therapy was based on implicated Candida species, sensitivity profile and prior treatments trialled. Fluconazole suppression was used most commonly (in 58% of patients), followed by nystatin in 47%, boric acid in 23%, clotrimazole in 14% and itraconazole in 7%. More than one type of antifungal was trialled in 37% of RVVC patients. Long-term antifungal combination therapy (³6 months) was initiated in 7% of cases. Off-license antifungals, including nystatin and boric acid pessaries, were required in 56% of women.
Overall the manuscript summarizes what a specialized facility has accomplished by following new RVVC guidelines. It seems that these guidelines could be followed in primary care and other healthcare facilities.